

# Pilot study of the relation between various dynamics of avatar experience and perceptual characteristics

Yuto Okamoto[1], Kazuhiro Matsui[1], Tetsuya Ando[1], Keita Atsuumi[1,2], Kazuhiro Taniguchi[1,3], Hiroaki Hirai[1] and Atsushi Nishikawa[1]

[1] Graduate School of Engineering Science, Osaka University, Toyonaka, Japan
[2] Graduate School of Information Sciences, Hiroshima City University, Hiroshima, Japan
[3] Faculty of Human Ecology, Yasuda Women's University, Hiroshima, Japan

Corresponding author
Kazuhiro Matsui,
matsui.kazuhiro.es@osaka-u.ac.jp

## ABSTRACT

In recent years, due to the prevalence of virtual reality (VR) and human-computer interaction (HCI) research, along with the expectation that understanding the process of establishing sense of ownership, sense of agency, and limb heaviness (in this study, limb heaviness is replaced with comfort level) will contribute to the development of various medical rehabilitation, various studies have been actively conducted in these fields. Previous studies have indicated that each perceptual characteristics decrease in response to positive delay. However, it is still unclear how each perceptual characteristic changes in response to negative delay. Therefore, the purpose of this study was to deduce how changes occur in the perceptual characteristics when certain settings are manipulated using the avatar developed in this study. This study conducted experiments using an avatar system developed for this research that uses electromyography as the interface. Two separate experiments involved twelve participants: a preliminary experiment and a main experiment. As observed in the previous study, it was confirmed that each perceptual characteristics decreased for positive delay. In addition, the range of the preliminary experiment was insufficient for the purpose of this study, which was to confirm the perceptual characteristics for negative delay, thus confirming the validity of conducting this experiment. Meanwhile, the main experiment showed that the sense of ownership, sense of agency, and comfort level decreased gradually as delay time decreased, (*i.e.*, this event is prior to action with intention, which could not be examined in the previous study). This suggests that control by the brain-machine interface is difficult to use when it is too fast. In addition, the distribution of the most strongly perceived settings in human perceptual characteristics was wider in regions with larger delays, suggesting this may lead to the evaluation of an internal model believed to exist in the human cerebellum. The avatar developed for this study may have the potential to create a new experimental paradigm for perceptual characteristics.

# INTRODUCTION

Sense of agency (SoA) is the sensation humans feel that they are causing physical changes in their own bodies and manipulated objects (*Haggard & Chambon, 2012*), and is said to be related to comfort (*Mishima et al., 2023*). In recent years, due to the prevalence of virtual reality (VR) and human–computer interaction (HCI) research, along with the expectation that understanding the process of establishing SoA will contribute to the development of rehabilitation (*e.g.*, contribution to the improvement of motor performance (*Matsumiya, 2021*) or to the renewal of body representations (*Wen et al., 2016*)), various studies have been actively conducted in these fields (*Haggard, 2008*). The widely used comparator model in which SoA is generated when the manipulated object behavior is close to the internal model that exists in the cerebellum, generates control input as a copy of one's own physical dynamics (*Wolpert, Ghahramani & Jordan, 1995*; *Blakemore, Frith & Wolpert, 2001*). There have been many reports that investigate how SoA is changed by generating and varying a positive delay in the reaction time of the manipulated object when it is manipulated (*Farrer, Valentin & Hupé, 2013*; *Wen, Yamashita & Asama, 2015a*; *Wen, Yamashita & Asama, 2015b*; *Osumi et al., 2019*; *Shimada, Qi & Hiraki, 2010*). For example, *Osumi et al. (2019)* used the computer graphic object which jumped by pushing a switch and in which delay to jumping was changed to test for SoA. This test includes an "event prior to action (EPA)" trial as a just control experiment (*i.e.*, the object was jumped prior to pushing the switch without relation to intention). The method of *Osumi et al. (2019 )* can only verify the effect of positive delay, but is unable to do so for negative delay. The occurrence of negative delay in previous studies is independent of "intention", making it impractical to conduct "intentional" EPA trials. A related study has investigated the correlation between positive delay and perceived characteristics, specifically the sense of ownership (SoO) and limb heaviness, distinct from SoA (*Osumi et al., 2018*; *Katayama et al., 2018*). Notably, SoO, which gauges "how much the avatar is perceived as equivalent to one's own body", is an important value when avatars are used for rehabilitation (*Kaneko, Yasojima & Kizuka, 2011*). On the other hand, limb heaviness is said to originate from the comparator model (*Katayama et al., 2018*). We interpreted this as pseudo-haptics (*Dominjon et al., 2005*; *Lécuyer, 2009*; *Issartel et al., 2015*; *Sarlegna, Baud-Bovy & Danion, 2010*; *Samad et al., 2019*; *Ujitoko & Ban, 2021*; *Flanagan & Beltzner, 2000*) occurring. Pseudo-haptics arises from the disparity between the internal model associated with SoA and the observed actual movement (*Flanagan & Beltzner, 2000*). The behind pseudo-haptics is akin to the generation of SoA. Therefore, herein, we have devised new method to ask not only the question of limb heaviness but also the question of lightness because pseudo-haptics of lightness occurs when an object in a virtual space moves faster, *i.e.*, negative delay, then the motion of a real object. Building on the knowledge that comfort is linked SoA (*Mishima et al., 2023*), we have redefined limb heaviness as comfort level (CL) and incorporated it into perceptual characteristics under investigation, there are precedents for the investigation of the relationship of these perceptual characteristics, SoA, SoO, and CL, to positive delay, as described above, but there are no precedents for the investigation of the relationship between these three types of perceptual characteristics and negative delay. Therefore, in this

pilot study, an avatar that uses electromyography (EMG) as an interface is proposed. The developed avatar can reflect the agonist–antagonist muscles ratio (AA ratio: equivalent to the equilibrium point, which is one of the human motor commands), which is the input, almost directly in the avatar (*Ando et al., 2023*). Since EMG activates prior to real body movement, it is possible to experience "an avatar that moves before the real body" which makes it possible to verify the "perceptual characteristics when the time delay is negative", (*i.e.,* this is EPA with intention which could not be achieved in previous reports). In addition to delay, overshoot characteristics and movement intensity can also be set. Thus, the developed avatar will enable various studies to investigate the relationship between motor control and perception as it explores the possibility of using avatars to evaluate new perceptual characteristics. The purpose of this study is to investigate SoA, SoO, and CL using this avatar system while participants experience an avatar featuring dynamics settings that induce negative delay–an aspect previously unexplored in existing reports. As a preliminary experiment to validate the avatar's basic performance employed settings similar to positive delay, confirming results akin to previous studies. Subsequently, we delved into the main experiment, investigating perceptual characteristics under negative delay as the main experiment.

## METHODS

### System

The avatar system developed by this study consists of a human elbow joint motion control dynamics model (Neuromusclularskeleton model: nMSS model) based on the equilibrium point hypothesis, which claims that the adjustment of two parameters, namely, the equilibrium point of the body and the joint stiffness, is important in human motion control (*Matsui et al., 2014*; *Matsui et al., 2015*; *Nagai et al., 2019*; *Matsui et al., 2022*). In this study, the equilibrium point movement is one-degree-of-freedom motion of the human elbow joint in the horizontal plane, which corresponds to the joint angle. AA ratio $r$ and agonist–antagonist muscles sum (AA sum) $s$, which are defined as the contributions to the joint stiffness, are expressed by Eqs. (1) and (2), where $m_e$ is the percent maximum voluntary contraction(%MVC)(*Vera-Garcia, Moreside & McGill, 2010*), which is normalized EMG obtained from EMG during maximal voluntary contraction in muscle alone, for the extensor side and $m_f$ is %MVC for the flexor side.

$$r = \frac{m_e}{m_f + m_e} \tag{1}$$

$$s = m_f + m_e \tag{2}$$

In this study, the NMSS model is applied to the human elbow joint control by EMG and is represented as a transfer function cascade coupling of the "neuromuscular system (second-order delay + dead time) + musculoskeletal system (second-order delay)" (Fig. 1), wherein each transfer function is expressed by Eqs. (3) and (4), where $G_{NM}(s)$, $K_{NM}$, $\omega_{nNM}$, $\zeta_{NM}$, and $L_{NM}$ are the transfer function, gain, natural angular frequency, damping coefficient, and dead time of the neuromuscular system, respectively, and $G_{MS}(s)$, $K_{MS}$,

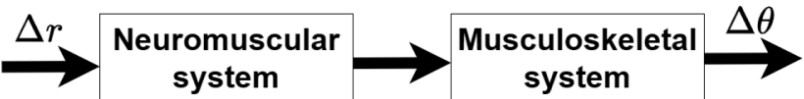

**Figure 1** NMSS model (quoted from *Matsui et al., 2015*), modified.

$\omega_{\mathrm{nMS}}$, $\zeta_{\mathrm{MS}}$, and $L_{\mathrm{MS}}$ are the transfer function, gain, natural angular frequency, damping coefficient, and dead time of the musculoskeletal system respectively.

$$G_{\mathrm{NM}}(s) = K_{\mathrm{NM}} \cdot \frac{\omega_{\mathrm{nNM}}^2}{s^2 + 2\zeta_{\mathrm{NM}}\omega_{\mathrm{nNM}}s + \omega_{\mathrm{nNM}}^2} \cdot e^{-L_{\mathrm{NM}}s} \tag{3}$$

$$G_{\mathrm{MS}}(s) = K_{\mathrm{MS}} \cdot \frac{\omega_{\mathrm{nMS}}^2}{s^2 + 2\zeta_{\mathrm{MS}}\omega_{\mathrm{nMS}}s + \omega_{\mathrm{nMS}}^2} \tag{4}$$

This avatar system can control virtual limbs "continuously and in real time" by utilizing NMSS model-based rather than EMG triggers (*Kaneko, 2016*). When using EMG as a trigger, a considerable time delay from the actual "intention" can occur based on the threshold setting. Furthermore, it only enables the realization of a constant avatar motion in accordance with the threshold value, lacking the capability to "intentionally stop joint motion at the intended position" like the NMSS model. Therefore, the NMSS model was adopted in this study. It has been shown that the human NMSS model parameters change according to the AA sum (*Iimura et al., 2011*; *Gong et al., 2020*; *Matsui et al., 2022*). Therefore, it is unlikely that humans sustain the AA sum $s$ constant during exercise, and when EMG is input to the NMSS model, it is desirable to sequentially change the model parameters according to the AA sum $s$ obtained from the EMG. However, for the sake of simplicity, at this stage, the NMSS model is regarded as a time-invariant model for the avatar system construction. In the experiment, EMGs ($m_{\mathrm{e}}$, $m_{\mathrm{f}}$) of muscles having opposing actions on the participant's right elbow joint are obtained, and the AA ratio $r$ expressed in Eq. (1) is input to the NMSS model in a PC to calculate the joint angle, which is reflected in an upper limb avatar placed on a VR with a head-mounted display (HMD, VIVE Pro, HTC Corporation). The avatar shoulder placement matches that of a real body captured by 3D motion capture Azure Kinect DK (Microsoft Corporation, Redmond, WA, USA).

The system configuration is shown in Fig. 2. In the experiment, the EMG of the extensor and flexor muscles, which are triceps brachii and biceps brachii, respectively, were obtained from electrode pads attached to the participant's upper arms, and after analog digital (AD) conversion, the data were processed in the PC. For the EMG acquisition, WEB-5000 (NIHON KOHDEN CORPORATION) was used in the preliminary experiments and EBM-102 (Unique Medical Co., Ltd., Tokyo, Japan) was used in the main experiments. Changes in equipment during the course of the study were due to changes in laws and regulations in Japan (*Telecommunications Bureau of the Ministry of Internal Affairs and Communications of Japan, 2005*). AD conversion and data acquisition were performed with

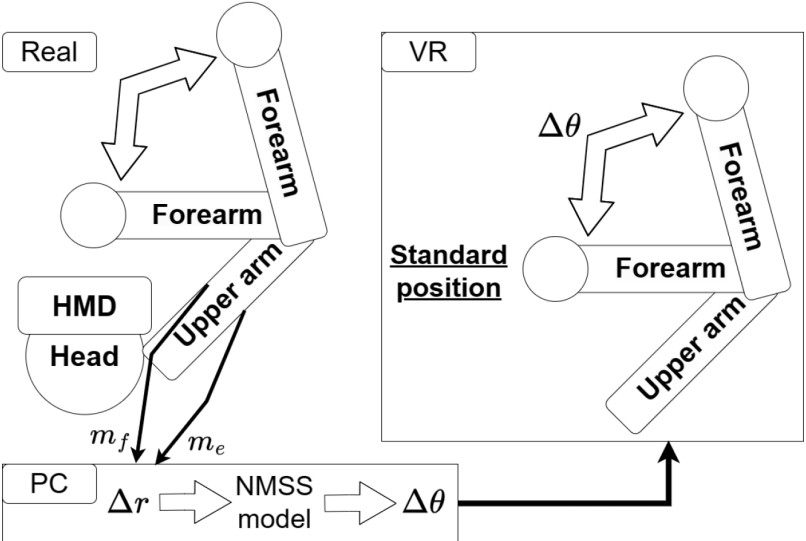

**Figure 2  System overview diagram.**

Labview2016 (NI) on a PC, combined with a data acquisition (DAQ) board. When using WEB-5000, EMG were acquired using a 10 Hz high-pass filter and a 100 Hz low-pass filter. Meanwhile, When using EBM-102, a 10 Hz high-pass filter and a 120 Hz low-pass filter ware used. Then, these filters were applied to the EMG acquired at a sampling frequency of 1000 Hz, and after rectification, a 22 Hz low-pass filter was applied and converted to %MVC ($m_e$, $m_f$).

The elbow joint angle calculated angular displacement $\Delta\theta(t)$ from the standard position is continuously reflected by 30 FPS on the upper limb elbow joint avatar in VR space in realtime. The standard position was set at 90°. Therefore, all trial experiments start from the real elbow joint at about 90°. With this system, the participant manipulates the virtual upper limb elbow joint avatar (Fig. 3) from the standard position using the EMG of the extensor and flexor muscles. The results are then visually displayed through VR. This means that the participant flexes and extends the virtual upper limb elbow joint avatar in VR space, which participants perceive their own arm with dynamics different from their respective physical dynamics (NMSS).

## Experiment

In this study, the experiment was conducted twice: preliminary experiment and main experiment.

The experiments involving participants in this study were conducted with the approval of the Ethics Committee on Research Involving Human Participants (R3-3-1) of the Graduate School of Engineering Science, Osaka University. Written informed consent was obtained from the participants.

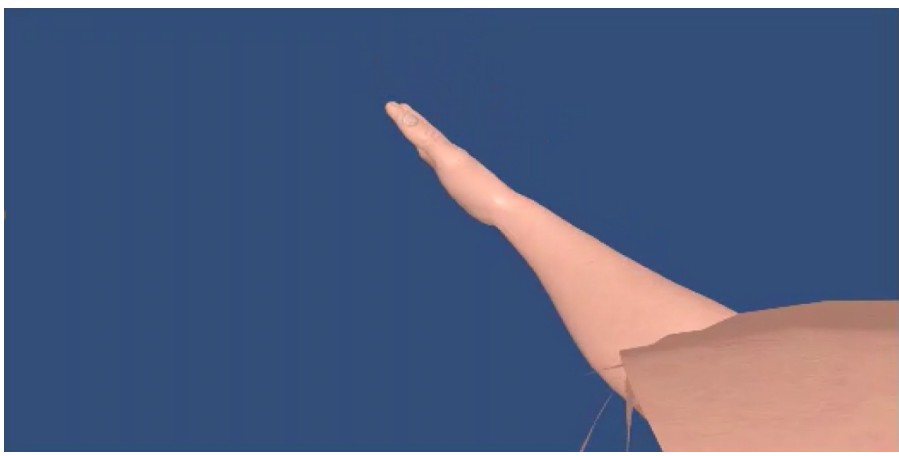

**Figure 3** Virtual upper limb avatar.

### Preliminary experiment

In a preliminary experiment, this study investigated how a change in SoO, SoA, and CL by manipulating avatar settings, specifically the natural angular frequency and dead time. In addition, discrimination threshold investigation was conducted, determining whether the avatar was perceived as displaying "lack of movement" or "hyper responsive movement" when settings were altered, details are provided in the Appendix. To validate the fundamental functions of the avatar, parameters unrelated to delay, such as gain and damping coefficient, were also examined and detailed in Appendix. For the research participants, six healthy participants (23.2 ± 0.7 years) wore HMD and experienced an upper limb avatar by performing the following TASK.

TASK: In VR space, the user rapidly repeats flexion and extension movements of the virtual upper limb elbow joint avatar five times in arbitrary timing at 90° and 150° which are remembered in VR space before TASK. If the user cannot flex the elbow at 90° or 150°, the user flexes it at the maximum or minimum angle that can be moved at that time.

Participants ware seated in a chair with their upper arm constrained to the platform and their forearm fixed to another platform which is movable in horizontal plane. Therefore, respective participants could move their elbow joint without any shoulder movement and gravitational pull. There were "comparison settings": natural angular frequency [rad/s] $(\omega_{npi} = \omega_{nNM} = \omega_{nMS})$ is { $\omega_{np1}, \omega_{np2}, \omega_{np3}, \omega_{np4}$} = {8, 6, 4, 2}, dead time [ms] $(L_i)$ is { $L_1, L_2, L_3, L_4$} = {0, 50, 150, 300}, and "standard setting": gain [−] $(K_{st} = K_{NM} \cdot K_{MS})$ is 300, natural angular frequency [rad/s] $(\omega_{npst} = \omega_{nNM} = \omega_{nMS})$ is 6, damping coefficient [−] $(\zeta_{st} = \zeta_{MN} = \zeta_{MS})$ is 0.7, dead time [ms] $(L_{st})$ is 0. Immediately after the standard setting a set of three comparison settings were experienced in sequence (in only final phase, two comparison settings were experienced). The comparison settings order was randomized so that participants experienced comparison settings twice. Comparison settings are combinations of gain, natural angular frequency, damping coefficient, and dead time, each of which varies from the standard setting while the other parameters are
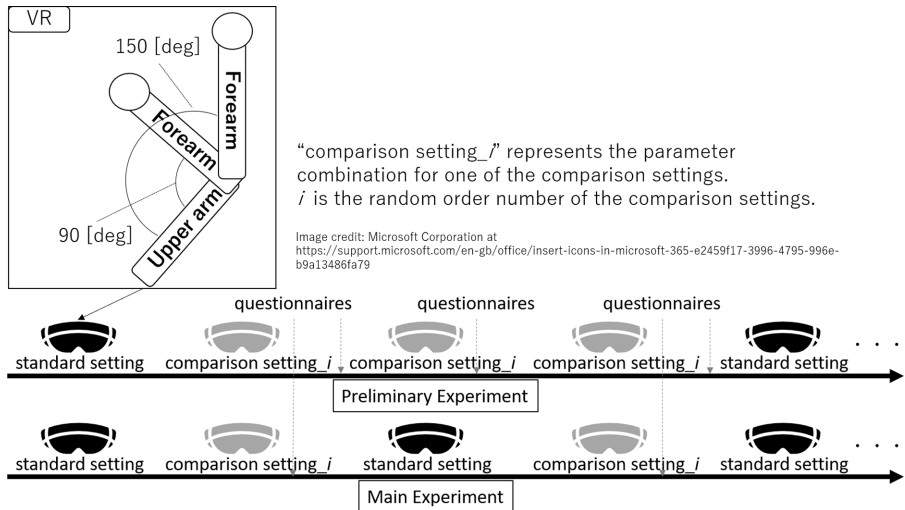

**Figure 4  Image illustrating that the experimental protocols in both the preliminary and main experiments, as well as the tasks, remained consistent between the standard and comparison settings in the VR space.**

constant. From this protocol, the dummy comparison settings is the same as the standard setting which occurs randomly.

After experiencing each of the comparison settings, participants answered each evaluation questionnaire orally while wearing the HMD. The following four questionnaires were administered. Three types of questionnaires were used to rate the "SoO (how much you felt the avatar is your own body)", the "SoA (how much you felt you were controlling the avatar yourself)", and the "CL (how much comfort you have when controlling the avatar)" on a worst to best scale of "−3 to +3" compared to the standard setting (0) in "0.25" increments. These experimental protocols is shown in Fig. 4.

### Main experiment

In the main experiment, the study focused only on natural frequency. Meanwhile, high natural frequency realizes the "avatar moves faster than the real body"; therefore, the study also investigated how a change in SoO, SoA, and, CL occur by this parameter. A different set of six healthy participants in participants (23.0 ± 1.9 years) HMD and experienced upper limb avatar by performing the same TASK as in the preliminary experiment. Participants were also seated in the same manner as the preliminary study group. However, while the standard setting is the same as that in the preliminary experiment, the following comparison settings differ from those in the preliminary experiment. In comparison settings, natural angular frequency [rad/s] ($\omega_{nmi} = \omega_{nNM} = \omega_{nMS}$) is { $\omega_{nm1}, \omega_{nm2}, \omega_{nm3}, \omega_{nm4}, \omega_{nm5}, \omega_{nm6}$} = {72, 36, 18, 9, 4, 2}. One of the comparison settings was experienced immediately after the standard setting. The comparison settings order was randomized so that each participant experienced each comparison setting twice. The comparison settings were applied so that they varied only with respect to the natural angular frequency, while the gain, damping coefficient, and dead time remained the same as those in the standard setting.

After experiencing each of the comparison settings, participants answered each evaluation questionnaire orally while wearing the HMD. The following three questionnaires were administered. Three types of questionnaires were used to rate SoO, SoA, and CL on worst to best a scale of " $-3$ to $+3$ " compared to the standard setting (0) in "0.25" increments. These experimental protocols is shown in Fig. 4.

### *Examples of system response*

Examples of the avatar system's responses to the NMSS model inputs are shown in Fig. 5. It also plots the output of the following sigmoidal function input for each of the comparison settings in the preliminary experiment and main experiment.

$$\Delta r = \frac{1}{1 + \exp(\frac{20}{3} \cdot (t - 0.5))} \tag{5}$$

where $\Delta r$ is an input that simulates the AA ratio with an initial value of 0 and $t$ denotes time. As indicated in the figure, natural angular frequency contributes to phase delay, and dead time contributes to time lag, respectively. Figure 5 plots the output of the actual AA ratio input *versus* the real elbow joint angle at that time in some main experiment comparison settings; as indicated in the figure, lower natural angular frequency contributes to negative delay and makes some small noise because AA ratio is directly reflected to the avatar joint angle. Settings other than the comparison settings shown in those legends follow the standard setting.

### *Analysis*

To examine differences between the comparison settings values in each questionnaire, averages of each values were compared by multiple comparisons (Holm–Bonferroni method with significance level 0.05). In this study, because sample size was smaller than those of previous studies (*Farrer, Valentin & Hupé, 2013*; *Wen, Yamashita & Asama, 2015a*; *Wen, Yamashita & Asama, 2015b*; *Osumi et al., 2019*), statistically using effect size was considered since it does not rely on sample size (*Field & Hole, 2003*). In order to clarify the relationship between the scales of difference, the $t$-value and the degrees of freedom ($df$) in the multiple comparisons were used to calculate effect size $ES$, which is expressed by the following equation:

$$ES = \sqrt{\frac{t^2}{t^2 + df}} \tag{6}$$

where effect size $ES$ is treated as follows from previous studies (*Osumi et al., 2019*): $ES \geq .5 (\ast\ast)$ indicates a "large" effect size, $ES \geq .3 (\ast)$ indicates a "medium" effect size, $ES \geq .1$ indicates a "small" effect size, and $ES \leq .1 (\ast)$ indicates a "slight" effect size. The statistical analysis software HAD (*Shimizu, 2016*) was used for the analysis.

In addition, the distributions of settings in which participants felt a strong SoO, SoA, and CL in the low-range natural angular frequency with some overlap were compared in preliminary and main experiments. Note that the accuracy of preliminary experiment, which is caused by their slightly differing protocols, is unsatisfied (*i.e.,* participants experience standard setting only one out of every three times). A setting which has a

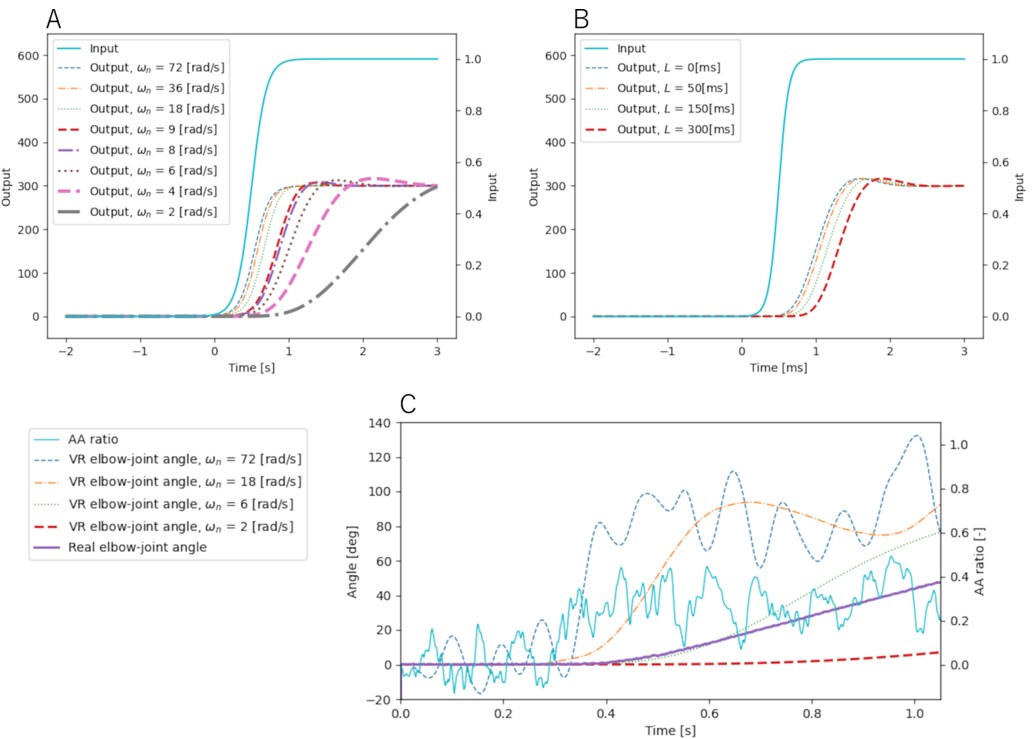

**Figure 5** Pattern of each setting—(A) natural angular frequency, (B) dead time, (C) real input and output.

maximum evaluated value is defined as "peak". If there were more than one setting which has a maximum evaluated value, all of these were defined as "peak". Distribution of these peaks was analyzed using histogram.

# RESULT

The values shown in each graph are the averaged value for all participants and its standard deviation. The values shown in the respective tables are the effect size of the multiple comparisons of the respective averaged values.

## Preliminary experiment
### Natural angular frequency

The evaluated values and effect sizes at natural angular frequency are shown in Fig. 6 and Table 1.

- SoO values peaked at $\omega_{np3}$ and fell at $\omega_{np4}$. The effect sizes of $\omega_{np1} - \omega_{np2}$ and $\omega_{np1} - \omega_{np4}$ were "medium", and all other effect sizes were "large".
- SoA values went down to $\omega_{np4}$. The effect sizes of $\omega_{np1} - \omega_{np4}$, $\omega_{np2} - \omega_{np4}$, and $\omega_{np3} - \omega_{np4}$ were "large".
- CL values showed a peak at $\omega_{np3}$ and a drop at $\omega_{np4}$. The effect size of $\omega_{np1} - \omega_{np3}$ was "medium", while that of $\omega_{np1} - \omega_{np4}$, $\omega_{np2} - \omega_{np3}$, $\omega_{np2} - \omega_{np4}$, and $\omega_{np3} - \omega_{np4}$ were "large".

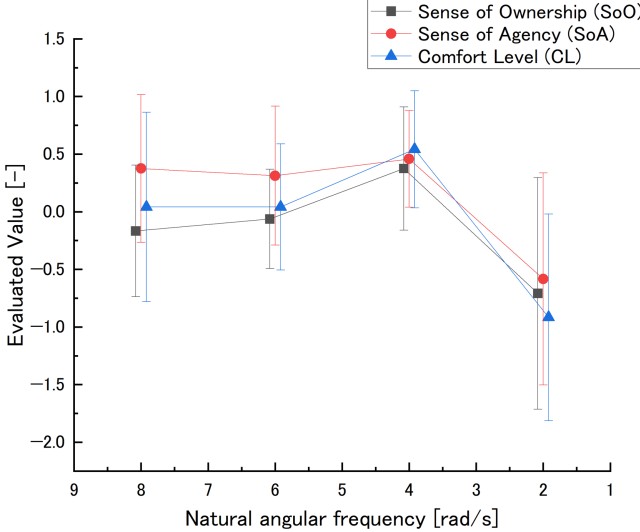

**Figure 6 Natural angular frequency in the preliminary experiment: evaluated value.** The natural angular frequency axis was reversed to be consistent with the dead time direction. The smaller the natural angular frequency, the larger the delay. However, because the natural angular frequency range was narrow, its axis is not log scale.

**Table 1 Natural angular frequency in the preliminary experiment: effect sizes (upper: SoO, middle: SoA, and lower: CL).**

|  | $\omega_{np2}$ | $\omega_{np3}$ | $\omega_{np4}$ |
|---|---|---|---|
| $\omega_{np1}$ | .38* | .70** | .41* |
| $\omega_{np2}$ |  | .64** | .51** |
| $\omega_{np3}$ |  |  | .73** |

|  | $\omega_{np2}$ | $\omega_{np3}$ | $\omega_{np4}$ |
|---|---|---|---|
| $\omega_{np1}$ | .07 | .15 | .62** |
| $\omega_{np2}$ |  | .27 | .68** |
| $\omega_{np3}$ |  |  | .72** |

|  | $\omega_{np2}$ | $\omega_{np3}$ | $\omega_{np4}$ |
|---|---|---|---|
| $\omega_{np1}$ | .00 | .40* | .53** |
| $\omega_{np2}$ |  | .61** | .70** |
| $\omega_{np3}$ |  |  | .86** |

## Dead time

The evaluated value and effect size at dead time are shown in Fig. 7 and Table 2.

- SoO values showed a small peak at $L_3$ and a drop at $L_4$. The effect sizes of $L_1 - L_3$ and $L_2 - L_3$ were "medium", while those of $L_1 - L_4$, $L_2 - L_4$, and $L_3 - L_4$ were "large".
- SoA values were lower at $L_4$. The effect sizes of $L_1 - L_4$, $L_2 - L_4$, and $L_3 - L_4$ were "large".

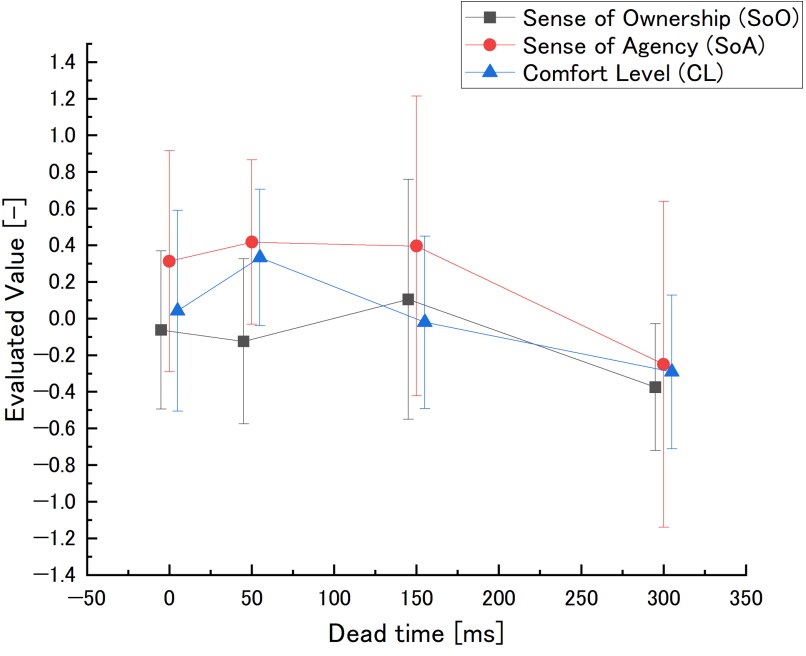

**Figure 7** Dead time in the preliminary experiment: evaluated value.

**Table 2** Dead time in the preliminary experiment: effect sizes (upper: SoO, middle: SoA, and lower: CL).

|       | $L_2$      | $L_3$      | $L_4$      |
|-------|-----------|-----------|-----------|
| $L_1$ | .13       | .38*      | .52**     |
| $L_2$ |           | .33*      | .52**     |
| $L_3$ |           |           | .58**     |

|       | $L_2$     | $L_3$     | $L_4$     |
|-------|-----------|-----------|-----------|
| $L_1$ | .13       | .15       | .69**     |
| $L_2$ |           | .02       | .67**     |
| $L_3$ |           |           | .67**     |

|       | $L_2$     | $L_3$     | $L_4$     |
|-------|-----------|-----------|-----------|
| $L_1$ | .79**     | .12       | .82**     |
| $L_2$ |           | .58**     | .96**     |
| $L_3$ |           |           | .43*      |

- CL values peaked at $L_2$ and decreased as the comparison setting value increased to $L_3$ and $L_4$. The effect size of $L_3 - L_4$ were "medium", while that of $L_1 - L_2$, $L_1 - L_4$, $L_2 - L_3$, and $L_2 - L_4$ were "large".

## Main experiment

The evaluated values and effect sizes at natural angular frequency are shown in Fig. 8 and Table 3.

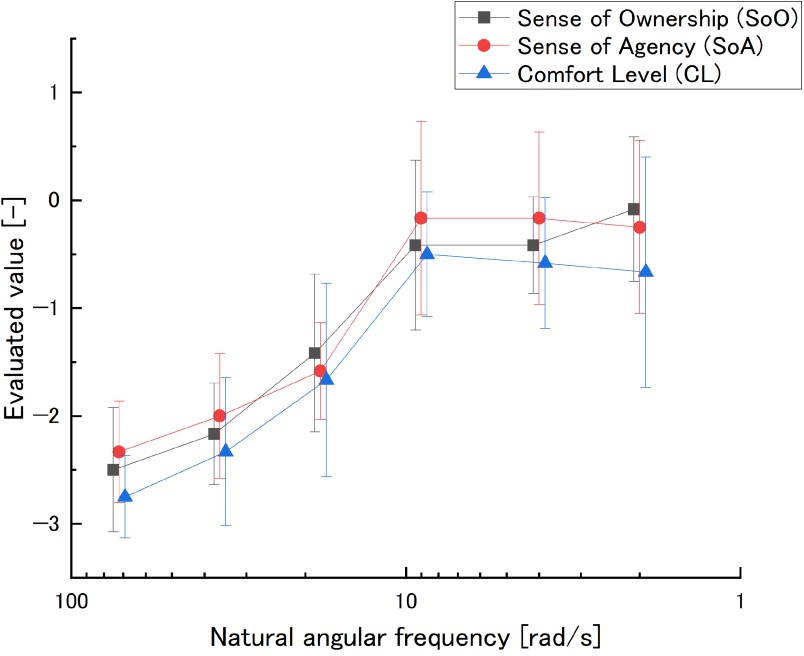

**Figure 8** **Natural angular frequency in the main experiment: evaluated value (semi-log plot).** The natural angular frequency axis was reversed to be consistent with the dead time direction. The smaller the natural angular frequency, the larger the delay.

- SoO values increased with decreasing values in the range $\omega_{nm1} \geq \omega_{nmi} \geq \omega_{nm4}$. Meanwhile, $\omega_{nm4} \geq \omega_{nmi} \geq \omega_{nm5}$ did not change much. $\omega_{nm5} \geq \omega_{nmi} \geq \omega_{nm6}$ increased with decreasing values. The value increased as the value decreased. The effect size of $\omega_{nm4} - \omega_{nm6}$ and $\omega_{nm5} - \omega_{nm6}$ showed "medium", all others showed "large".
- SoA values increased with decreasing values in the range $\omega_{nm1} \geq \omega_{nmi} \geq \omega_{nm4}$. Meanwhile, $\omega_{nm4} \geq \omega_{nmi} \geq \omega_{nm5}$, did not change much. $\omega_{nm5} \geq \omega_{nmi} \geq \omega_{nm6}$ decreased with decreasing values. The values decreased with decreasing values. All effect sizes except $\omega_{nm4} - \omega_{nm5}$, $\omega_{nm4} - \omega_{nm6}$, and $\omega_{nm5} - \omega_{nm6}$ showed "large".
- CL values increased with decreasing value in the range $\omega_{nm1} \geq \omega_{nmi} \geq \omega_{nm4}$ and decreased with decreasing value in the range $\omega_{nm4} \geq \omega_{nmi} \geq \omega_{nm6}$. In other words, it peaked at $\omega_{nm4}$. All effect sizes except $\omega_{nm4} - \omega_{nm5}$, $\omega_{nm4} - \omega_{nm6}$, and $\omega_{nm5} - \omega_{nm6}$ showed "large".

## Distribution in the low natural angular frequency domain

Figure 9 shows the distribution of peak evaluations of natural angular frequency in the low range for each research participant. It can be seen that the preliminary experiment peak is closer to the high frequency region compared to the main experiment. In other words, in this experiment, the low frequency region distribution of the natural angular frequency changed.
**Table 3  Natural angular frequency in the main experiment: effect sizes (upper: SoO, middle: SoA, and lower: CL).**

|  | $\omega_{nm2}$ | $\omega_{nm3}$ | $\omega_{nm4}$ | $\omega_{nm5}$ | $\omega_{nm6}$ |
|---|---|---|---|---|---|
| $\omega_{nm1}$ | .67** | .90** | .91** | .92** | .99** |
| $\omega_{nm2}$ |  | .80** | .86** | .92** | .97** |
| $\omega_{nm3}$ |  |  | .69** | .77** | .84** |
| $\omega_{nm4}$ |  |  |  | .00 | .31* |
| $\omega_{nm5}$ |  |  |  |  | .30* |

|  | $\omega_{nm2}$ | $\omega_{nm3}$ | $\omega_{nm4}$ | $\omega_{nm5}$ | $\omega_{nm6}$ |
|---|---|---|---|---|---|
| $\omega_{nm1}$ | .58** | .84** | .92** | .90** | .98** |
| $\omega_{nm2}$ |  | .62** | .94** | .86** | .93** |
| $\omega_{nm3}$ |  |  | .82** | .85** | .84** |
| $\omega_{nm4}$ |  |  |  | .00 | .07 |
| $\omega_{nm5}$ |  |  |  |  | .06 |

|  | $\omega_{nm2}$ | $\omega_{nm3}$ | $\omega_{nm4}$ | $\omega_{nm5}$ | $\omega_{nm6}$ |
|---|---|---|---|---|---|
| $\omega_{nm1}$ | .68** | .81** | .98** | .96** | .94** |
| $\omega_{nm2}$ |  | .70** | .96** | .93** | .90** |
| $\omega_{nm3}$ |  |  | .86** | .83** | .65** |
| $\omega_{nm4}$ |  |  |  | .11 | .15 |
| $\omega_{nm5}$ |  |  |  |  | .08 |

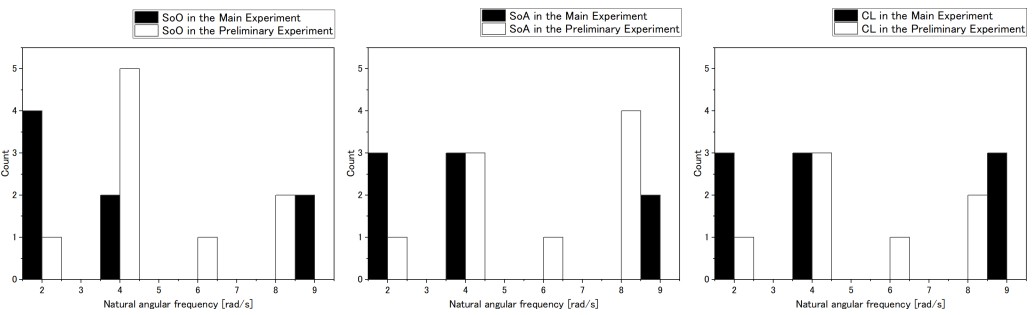

**Figure 9  Histograms of the low natural angular frequency domain.**

# DISCUSSION

## Preliminary experiment

### Natural angular frequency

SoO peaked at $\omega_{np3}$ and fell at $\omega_{np4}$. The effect sizes of $\omega_{np1} - \omega_{np2}$, $\omega_{np2} - \omega_{np3}$, and $\omega_{np3} - \omega_{np4}$, which compare adjacent comparison settings, respectively show "large", "large", and "medium". Meanwhile, $\omega_{np3}$ has "large" effect to $\omega_{np1}$, $\omega_{np2}$ and $\omega_{np4}$, which indicates $\omega_{np3}$ is peak. Furthermore, $\omega_{np4}$ has "large" effect to $\omega_{np3}$, $\omega_{np2}$, which indicates $\omega_{np4}$ is bottom. In addition, $\omega_{np4}$ has "medium" effect to $\omega_{np1}$ and $\omega_{np3}$ has "large" effect

to $\omega_{np1}$, which indicates SoO gradually decreases by increasing natural angular frequency, *i.e.*, $\omega_{np3}$ is peak.

SoA decreased at $\omega_{np4}$. However, looking at the effect size of $\omega_{np1} - \omega_{np2}$, $\omega_n2 - \omega_{np3}$, and $\omega_{np3} - \omega_{np4}$, which compare adjacent comparison settings, respectively shows "large", "small", and "slight". Meanwhile, $\omega_{np4}$ has "large" effect to all other settings. Therefore, SoA descended at $\omega_{np4}$ and SoA's peak is not clear.

CL showed an increase and peak at $\omega_{np3}$ and a decrease at $\omega_{np4}$. In addition, looking at the effect sizes of $\omega_{np1} - \omega_{np2}$, $\omega_{np2} - \omega_{np3}$, and $\omega_{np3} - \omega_{np4}$, which compare adjacent comparison settings, respectively show "slight", "large", and "large". Meanwhile, $\omega_{np3}$ has "large" effect to $\omega_{np2}$ and $\omega_{np4}$ has "medium" effect to $\omega_{np1}$, which indicates $\omega_{np3}$ is peak. Furthermore, $\omega_{np4}$ has "large" effect to all other settings, which indicates $\omega_{np4}$ is bottom. In addition, $\omega_{np4}$ has "large" effect to $\omega_{np1}$ and $\omega_{np3}$ has "medium" effect to $\omega_{np1}$, which indicates CL more gradually decreases by increasing natural angular frequency than SoO.

Natural angular frequency is a parameter of phase delay which is dominant when there is no time lag. These results indicate that a low natural angular frequency, which causes a larger delay, make SoO, SoA, and CL decrease. The similar results as in the previous study are considered to have been obtained. A high natural angular frequency, which causes a smaller delay, has the possibility to make SoO and CL decrease similar to that in large delay. These results suggest that additional experiments that have more high natural angular frequency are necessary. In addition, these results also indicate that there is a possibility that CL has a more acute index than SoA because CL had the peak.

### Dead time

SoO showed a small peak at $L_3$ and a drop at $L_4$. The effect sizes of $L_1 - L_2$, $L_2 - L_3$, and $L_3 - L_4$, which compare adjacent comparison settings, respectively, show "small", "medium", and "large". Meanwhile, $L_3$ has "medium" effect to $L_1$, $L_2$, which indicates $L_3$ is the peak. Furthermore, $L_4$ has "large" effect to all other settings, which indicates $L_4$ is the bottom. Therefore, it can be said that the downward trend at $L_4$ and peak of $L_3$ is effective.

SoA went down at $L_4$. The effect size of $L_1 - L_2$, $L_2 - L_3$, and $L_3 - L_4$, which compare adjacent comparison settings, respectively show "small", "slight", and "large". Meanwhile, L4 has "large" effect to all other settings, which indicates L4 is the bottom. Therefore, the downward trend at $L_4$ is effective.

CL peaked at $L_2$, and then decreased as the comparison setting value increased to $L_3$ and $L_4$. The effect sizes of $L_1 - L_2$, $L_2 - L_3$, and $L_3 - L_4$, which compare adjacent comparison settings, respectively, show "large", "large", and "medium".

Dead time is a parameter of time lag. These results indicate that a high dead time, which causes a larger time lag, make SoO, SoA, and CL decrease. The trend of SoA is similar to that of the previous study (*Osumi et al., 2019*). About the peak of $L_2$ in CL, it is considered that because the avatar standard setting moves faster than the real body to not feel comfort, $L_2$ in which the avatar is delayed is the peak. Furthermore, the peak of SoO is $L_3$, which is different from CL. This indicates that it has the possibility that the trend of generating SoO in time lag is different from SoA and CL.

### SoA at dummy setting

In natural angular frequency and dead time, SoA was higher than 0 in the dummy setting, which is the same case as in the standard setting. This suggests that SoA may be improved by repetition compared to the other perceptual characteristics, since the participants always experience the avatar in each setting after experiencing the standard setting due to the experimental protocol. In contrast, CL is close to 0 in the dummy setting. CL is introduced as an alternative parameter to limb heaviness to encompass inquiries not only about the sensation of heaviness but also about the sensation of lightness. This additional information by reports suggest a relation between comfort and SoA (*Mishima et al., 2023*). SoA is changed from this result and CL may be a more acute questionnaire method in some cases.

### Summary of preliminary experiment

In the preliminary experiment, we verified that similar results to those observed in the previous study could be replicated for positive delays using the settings of natural angular frequency and dead time. Simultaneously, we gained new insights into relationship between each parameter and perception through this avatar experience system. Moreover, we affirmed the validity of main experiment, as the setting range explored in the preliminary experiment was deemed insufficient for confirming the perceptual characteristics in negative delay, which is the purpose of this study.

### Main experiment

SoO increased with decreasing natural angular frequency values in the range of $\omega_{nm1} \geq \omega_{nmi} \geq \omega_{nm4}$. SoO did not change much in the range of $\omega_{nm4} \geq \omega_{nmi} \geq \omega_{nm5}$. In the range of $\omega_{nm5} \geq \omega_{nmi} \geq \omega_{nm6}$, SoO increased with a decreasing natural angular frequency value. In addition, looking at the effect sizes for $\omega_{nm1} - \omega_{nm2}$, $\omega_{nm2} - \omega_{nm3}$, $\omega_{nm3} - \omega_{nm4}$, $\omega_{nm4} - \omega_{nm5}$, and $\omega_{nm5} - \omega_{nm6}$, which compare adjacent comparison settings, respectively show "large", "large", "large", "slight", and "medium". Meanwhile, $\omega_{nm4}$ has "medium" effect to $\omega_{nm6}$.

SoA increased with a decreasing natural angular frequency value in the range of $\omega_{nm1} \geq \omega_{nmi} \geq \omega_{nm4}$. SoA did not change much in the range of $\omega_{nm4} \geq \omega_{nmi} \geq \omega_{nm5}$. In the range of $\omega_{nm5} \geq \omega_{nmi} \geq \omega_{nm6}$, SoA decreased as the natural angular frequency value decreased. Looking at the effect size of $\omega_{nm1} - \omega_{nm2}$, $\omega_{nm2} - \omega_{n3}$, $\omega_{nm3} - \omega_{nm4}$, $\omega_{nm4} - \omega_{nm5}$, and $\omega_{nm5} - \omega_{nm6}$, which compare adjacent comparison settings, respectively show "large", "large", "large", "slight", and "slight". Meanwhile, $\omega_{n4}$ has "medium" effect to $\omega_{nm6}$. Thus, the downward trend of SoA in the range of $\omega_{nm4} \geq \omega_{nmi} \geq \omega_{nm6}$ is hardly effective. However, in the range of $\omega_{nm1} \geq \omega_{nmi} \geq \omega_{nm4}$, the tendency of SoA to increase is effective.

CL increased with a decreasing natural angular frequency value in the range of $\omega_{nm1} \geq \omega_{nmi} \geq \omega_{nm4}$ and decreased with a decreasing value in the range of $\omega_{nm4} \geq \omega_{nmi} \geq \omega_{nm6}$. Looking at the effect sizes for $\omega_{nm1} - \omega_{nm2}$, $\omega_{nm2} - \omega_{nm3}$, $\omega_{nm3} - \omega_{nm4}$, $\omega_{n4} - \omega_{nm5}$, and $\omega_{nm5} - \omega_{nm6}$, which compare adjacent comparison settings, respectively show "large", "large", "large", "small" and "slight". Thus, in the range of $\omega_{nm4} \geq \omega_{nmi} \geq \omega_{n6}$, the

downward trend of CL is hardly effective. However, in the range $\omega_{nm1} \geq \omega_{nmi} \geq \omega_{nm4}$, the upward trend of CL is effective.

This experiment is an additional experiment based on the preliminary experiment results. As noted earlier, natural angular frequency is a parameter of delay. These results indicate that a high natural angular frequency, which causes a smaller delay or negative delay, has the possibility to makes SoO, SoA, and CL decrease similar to a large delay. In a previous study, EPA trial without "intention" was only discussed in that it makes no SoA. In this study, it is indicated that SoA decreases gradually by decreasing delay. This delay has the possibility to be negative and is based on voluntary signal. Therefore, this situation can be called EPA trial with "intention". Meanwhile, CL and SoO have a similar trend with SoA. Since increasing natural angular frequency makes some small noise, this point has to be investigated by using avatar with intentional noise and not rely on natural angular frequency in future studies. In contrast, a low natural angular frequency area similar to that in the preliminary experiment has a different trend than that in the preliminary experiment, *i.e.,* there is no decreasing trend at $L_4$. It is considered that it is caused by distribution of perceptual characteristic distribution for each participant.

### Distribution in the low natural angular frequency domain

The histograms of Fig. 9 show that the distribution of the peaks in the evaluation at natural angular frequencies is different for each individual. This suggests that peaks in the low frequency range are different for each individual. These results might mean that the natural angular frequency value appropriate for each individual is different, and those dynamics around these peaks are close to those corresponding to the internal model of each individual participant. Hence, the histograms of the peaks in the preliminary experiment and main experiment suggest the potential for personalized peaks unique to each individual. This observation opens the possibility for evaluating an internal model that has not been realized thus far.

## CONCLUSION

This pilot study was conducted to explore the possibility of using the developed avatar system to evaluate new perceptual characteristics. Various natural angular frequency and dead time were investigated. The study conducted questionnaire evaluations of SoO, SoA, and CL.

From the results of the preliminary experiments, the following points were indicated. Meanwhile, in the natural angular frequency parameter, a larger delay makes SoO, SoA, and CL decrease. In the dead time parameter, results indicate that a larger time lag makes SoO, SoA, and CL decrease and find some peaks, which indicate that it has the possibility that the trend of generating SoO in time lag is different from SoA and CL. Furthermore, it was indicated that CL may be a more acute questionnaire method than SoA in some cases. In conclusion, the preliminary experiments yielded results similar to those observed in previous studies, underscoring the necessity for main experiment with broader setting range.

From the results of the main experiments, the EPA trial with "intention" revealed that SoO, SoA, and CL decreases gradually by decreasing delay. This idea leads to the suggestion for a brain–machine interface, *i.e.,* recently fast detection BMI is in development (*Asai et al., 2022*), however, there is a possibility that too fast a control by BMI is difficult to use. In addition, the larger delay regions showed a wider distribution of the most strongly perceived settings in human perceptual properties, suggesting that this may lead to an evaluation of the internal model that is supposed to exist in the human cerebellum (*Miall & Wolpert, 1996*). Furthermore, the characteristics of changes in the SoO suggest a guideline for how avatars should behave to improve immersion in rehabilitation and general entertainment applications. Finally, through this study it has been shown that it may be possible to develop a new experimental paradigm for perceptual properties using the study developed avatars.

Based on the findings of this research, determining variations in setting parameters and the ideal sample size are necessary for a more detailed future study. This present study provides guidance on the further development of rehabilitation equipment, avatar interfaces for VR, and robot control interfaces.

**Abbreviations**

| | |
|---|---|
| **AA ratio** | Agonist–antagonist muscles ratio. The ratio of the muscle activity among opposing muscles is determined from the %MVC of EMG of the flexor and extensor muscles as described in previous studies (*Iimura et al., 2011*; *Gong et al., 2020*; *Matsui et al., 2022*). This is used to manipulate the avatar developed in this study (*Ando et al., 2023*). |
| **AA sum** | Agonist–antagonist muscles sum. The total muscle activity among the opposing muscles determined from the %MVC of EMG of flexor and extensor muscles, as described in previous studies (*Iimura et al., 2011*; *Gong et al., 2020*; *Matsui et al., 2022*). While the original human believed to control AA sum during movements, this study did not utilize it as a control variable in the developed avatar. |
| **CL** | Comfort level. A newly defined index introduced in this study serves as a replacement for the perceived characteristic of limb heaviness, previously compared to positive delay in previous studies. (*Osumi et al., 2019*; *Osumi et al., 2018*; *Katayama et al., 2018*). The question is asked as "how much comfort you have when controlling the avatar". |
| **EMG** | Electromyography. A method of acquiring muscle activity as voltage. |
| **EPA** | Event prior action. An action occurring before the user's physical action, as "the object jumps before the button is pressed." Although previous studies, has been verified, but EPA "with intention" has not been verified yet. Through the usage of the avatar developed in this study, users can experience "a body that moves before their own actual body" using EMGs, which reflect "their intentions." Thus, our avatar allows for an EPA with "intention" in this study. |
| **HCI** | Human–computer interaction. This indicates general human–computer interaction. |

| | |
|---|---|
| **HMD** | Head-mounted display. Goggle-type VR presentation device. |
| **NMSS model** | Neuromusculoskeletal model. A mathematical model for calculating avatar joint angles that is based on previous studies (*Matsui et al., 2014*; *Matsui et al., 2015*; *Nagai et al., 2019*; *Matsui et al., 2022*). |
| **SoA** | Sense of agency. A commonly used measure to assess whether individuals feel they are actively moving object themselves is framed as "how much you felt you were controlling the avatar yourself". |
| **SoO** | Sense of ownership. A commonly used measure of whether the avatar is person's own body. The question is asked as "how much you felt the avatar is your own body". |
| **VR** | Virtual reality. A virtual space that is different from the real world |
| **%MVC** | Percent maximum voluntary contraction. Commonly used normalize method of electromyography (*Vera-Garcia, Moreside & McGill, 2010*). |

### Funding
This work was funded by the JSPS KAKENHI JP20K14693, the Future Social Value Co-Creation Project, Osaka University. The funders had no role in study design, data collection and analysis, decision to publish, or preparation of the manuscript.

### Grant Disclosures
The following grant information was disclosed by the authors:
The JSPS KAKENHI JP20K14693.
The Future Social Value Co-Creation Project, Osaka University.

### Competing Interests
The authors declare there are no competing interests.

### Author Contributions

- Yuto Okamoto conceived and designed the experiments, performed the experiments, analyzed the data, performed the computation work, prepared figures and/or tables, authored or reviewed drafts of the article, and approved the final draft.
- Kazuhiro Matsui conceived and designed the experiments, performed the experiments, analyzed the data, performed the computation work, prepared figures and/or tables, authored or reviewed drafts of the article, and approved the final draft.
- Tetsuya Ando conceived and designed the experiments, performed the experiments, performed the computation work, prepared figures and/or tables, authored or reviewed drafts of the article, and approved the final draft.
- Keita Atsuumi conceived and designed the experiments, authored or reviewed drafts of the article, and approved the final draft.
- Kazuhiro Taniguchi conceived and designed the experiments, authored or reviewed drafts of the article, and approved the final draft.

- Hiroaki Hirai conceived and designed the experiments, authored or reviewed drafts of the article, and approved the final draft.
- Atsushi Nishikawa conceived and designed the experiments, authored or reviewed drafts of the article, and approved the final draft.

## Data Availability

The raw measurements are available in the Supplementary File.

## Supplemental Information

Supplemental information for this article can be found online at http://dx.doi.org/10.7717/peerj-cs.2042#supplemental-information.

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
