# Peer review of "Pilot study of the relation between various dynamics of avatar experience and perceptual characteristics"

_PeerJ Computer Science, doi:10.7717/peerj-cs.2042_

## Round 0.1 · original submission · Major Revisions

Please revise the manuscript according to the reviews.

Reviewer 1 ·

Basic reporting

The proposed study investigate the discrimination thresholds of dynamics parameters and
deduce how changes occur in the senses of ownership and agency when certain settings
are manipulated using the avatar developed in this study. This study conducted
experiments using an avatar system developed for this research that uses
electromyography as the interface. Two separate experiments involved twelve
participants: a preliminary experiment and a main experiment. The preliminary
experiment showed that discrimination for gain, natural angular frequency, and damping
coeûcient can be achieved with minimal resolution. However, discrimination for dead time
suggested that illusion is perceived, but dead time itself cannot be discriminated.
Furthermore, perceptual characteristics for avatar setting parameters were conûrmed in
the preliminary experiment. Meanwhile, the main experiment showed that the sense of
ownership, sense of agency, and comfort level decreased gradually as delay time
decreased, (i.e., this event is prior to action with intention, which could not be examined in
the previous study). This suggests that control by the brain3machine interface is diûcult to
use when it is too fast. In addition, the distribution of the most strongly perceived settings
in human perceptual characteristics was wider in regions with larger delays, suggesting
this may lead to the evaluation of an internal model believed to exist in the human
cerebellum.
The introduction does not reflect the actual understanding and contribution of the proposed study. Additionally, the abstract falls short of representing the actual contribution.

Why is there a need for dual experiments such as preliminary experiments and the actual main experiment? Furthermore, a large number of abbreviations are causing difficulties in the introduction.

There is no related work to demonstrate the distinction of this study. The introduction does not present enough information about the background, etc.

The results seem to be over-optimized, suggesting significant optimality in the proposed work. The overall organization of the article is not well-written, and the structure of the article needs major improvements.

What is gained in preliminary experiments? How did the authors utilize them?

I suggest the authors include only those results that are crucial for this study.

Experimental design

Included in basic reporting.

Validity of the findings

Included in basic reporting.

·

Basic reporting

The paper titled "Pilot study of the relation between various dynamics of avatar experience and perceptual characteristics", seems very interesting research. The purpose of this study was to investigate the discrimination thresholds of dynamics parameters and deduce how changes occur in the senses of ownership and agency when certain settings are manipulated using the avatar developed in this study This research has done satisfactory literature study to find the research objective and motivation for this paper. Therefore, in this pilot study, an avatar that uses electromyography (EMG) as an interface is proposed. The avatar system developed by this study consists of a human elbow joint motion control dynamics model (Neuro-musclularskeletal model: NMSS model). The paper is clearly written in a good style. Some latest research work needs to be cited

Experimental design

The objective and motivation for the research has been very well stated in the introduction part. But needs clarification on the following:
1. Methodology of this pilot study require more explanation to understand the intent of author
2. In this study authors have chosen NMSS Model than EMG Triggers. The motivation for NMSS Model should need clarification.

Validity of the findings

The authors adequately evaluated their work, and all claims are clearly articulated and supported by empirical experiments.

Additional comments

This research paper is very interesting study. Problem statement is formulated in a very significant way. The proposed methodology should be more objective centric.

---

## Round 0.2 · accepted · Accept

The authors carefully addressed the comments from the reviewers.

Reviewer 1 ·

Basic reporting

The authors have addressed all previous concerns.

Experimental design

NA

Validity of the findings

NA

·

Basic reporting

The article "Pilot study of the relation between various dynamics of avatar experience and perceptual characteristics" seems novel and timely.

Experimental design

A Detailed experimental analysis has been done to validate the proposed approach.

Validity of the findings

This pilot study was conducted to explore the possibility of using the developed avatar system to evaluate
new perceptual characteristics with sufficient details and literature analysis.

Additional comments

The article has improved and updated sufficiently.